# Urinary tract infections, risk factors and antimicrobial resistance patterns in heart failure patients on sodium-glucose transporter 2 inhibitors: Evidence from Jakaya Kikwete Cardiac Institute in Tanzania

**Julieth Kaaya[1], Martine A. Manguzu[2], Deus Buma[3], Hamu Mlyuka[2], Samwel Rweyemamu[4], Albert Ntukula[5], Helfrid B. Ilomo[3], Elias Bukundi[6], Raphael Sangeda[7], Ritah F. Mutagonda[2]***

1 Department of Pharmacy, Muhimbili Orthopedic Institute, Dar es Salaam, Tanzania, 2 Department of Clinical Pharmacy and Pharmacology, Muhimbili University of Health and Allied Sciences, Dar es Salaam, Tanzania, 3 Department of Pharmacy, Muhimbili National Hospital, Dar es Salaam, Tanzania, 4 Jakaya Kikwete Cardiac Institute, Dar es Salaam, Tanzania, 5 Department of Microbiology, Central Pathology Laboratory, Muhimbili National Hospital, Dar es Salaam, Tanzania, 6 Department of Epidemiology and Biostatistics, School of Public Health and Social Sciences, Muhimbili University of Health and Allied Sciences, Dar es Salaam, Tanzania, 7 Department of Pharmaceutical Microbiology, School of Pharmacy, Muhimbili University of Health and Allied Sciences, Dar es Salaam, Tanzania

* rittdavisrida@yahoo.com

## Abstract

### Background

Sodium-glucose co-transporter 2 (SGLT2) inhibitors such as dapagliflozin and empagliflozin are increasingly used in heart failure (HF) management due to their cardiovascular benefits. However, glycosuria induced by SGLT2 inhibition may increase the risk of urinary tract infections (UTIs). Limited data exist on UTI burden among HF patients on SGLT2 inhibitors in low-resource settings.

### Objective

This study aimed to determine the prevalence of UTIs, associated factors, and antimicrobial susceptibility patterns of uropathogens in HF patients receiving SGLT2 inhibitors at the Jakaya Kikwete Cardiac Institute (JKCI) in Tanzania.

### Methods

A hospital-based cross-sectional study was conducted from March to June 2024 among HF patients aged ≥18 years on SGLT2 inhibitors. Data was collected using structured questionnaires and medical records. Midstream clean-catch urine samples were collected in sterile containers and processed using semi-quantitative urine culture on CLED and blood agar and subjected to antimicrobial susceptibility

**Data availability statement:** All relevant data are within the paper and its Supporting Information files.

**Funding:** The author(s) received no specific funding for this work.

**Competing interests:** The authors have declared that no competing interests exist.

**List of Abbreviations:** ADR: Adverse Drug Reaction; AMR: Antimicrobial Resistance; CFU: Colony Forming Unit; CLSI: Clinical and Laboratory Standards Institute; CPL: Central Pathology Laboratory; HF: Heart failure; HPF: High power field; JKCI: Jakaya Kikwete Cardiac Institute; NYHA: New York Heart Association; RCTs: Randomized Clinical Trials; SGLT2i: Sodium-glucose cotransporter 2 Inhibitor; TMDA: Tanzania Medicines and Medical Devices Authority; UPEC: Uropathogenic Escherichia coli; UTI: Urinary Tract Infections.

testing using the Kirby-Bauer disk diffusion method per CLSI M100 (2024) guidelines. Descriptive statistics and modified Poisson regression were used for analysis.

## Results

Out of 138 urine samples processed, 22 (15.9%) showed significant growth. The most common uropathogen was *Escherichia coli* (50.0%), followed by *Klebsiella pneumoniae* (13.6%), *Pseudomonas aeruginosa* (9.1%), and *Candida* spp. (9.1%). Significant risk factors for UTI included age > 60 years (aPR 3.77; 95% CI: 1.42–10.01), female sex (aPR 2.92; 95% CI: 1.19–7.15), and SGLT2 inhibitor use ≥ 4 months (aPR 3.19; 95% CI: 1.70–5.96). High resistance was observed among bacterial isolates against ampicillin (100%), tetracycline (69.2%), and ceftazidime (53.8%), whereas high susceptibility was noted against nitrofurantoin (84.6%) and meropenem (100%).

## Conclusion

UTIs are common among HF patients on SGLT2 inhibitors, with E. coli as the predominant pathogen and a concerning resistance to commonly used antibiotics. These findings underscore the need for routine urine culture and sensitivity testing to guide appropriate therapy and promote antimicrobial stewardship, particularly in resource-constrained settings.

## Background

Sodium-glucose co-transporters (SGLT2) inhibitors, including Dapagliflozin, Empagliflozin, Canagliflozin, and Sotagliflozin, are a class of oral antihyperglycemic agents that lower blood glucose by inhibiting its reabsorption in the proximal tubule [1]. In addition to their glucose-lowering effect, SGLT2 inhibitors have demonstrated significant cardiovascular and renal benefits, including reductions in blood pressure, hospitalization for heart failure, and cardiovascular mortality [2]. These findings have led to expanded use beyond diabetes, particularly in the management of heart failure with reduced ejection fraction (HFrEF), where they are now recommended as part of guideline-directed medical therapy [3].

Despite these therapeutic advantages, SGLT2 inhibitors have been associated with an increased risk of genitourinary infections [4]. The mechanism is attributed to induced persistent glucosuria, which creates a glucose-rich urinary environment that may facilitate microbial growth and proliferation [1,4]. This issue warrants particular attention among patients with heart failure, who frequently possess additional predisposing factors for UTI, including comorbidities, impaired immune function, and frequent hospitalization [5]. Furthermore, UTIs in individuals with HF can exacerbate clinical symptoms, precipitate decompensation, and prolong hospitalization, highlighting the need for timely diagnosis and prevention [6]

While the link between SGLT2 inhibitors and genital mycotic infections is well established, evidence regarding their association with urinary tract infections (UTIs)

remains inconsistent. For example, in DAPA-HF trials, UTI rates were generally low and similar between the treatment groups, and this was because UTI was not labeled as an important safety concern event; thus, events may have been less reported [7]. Yet other studies have described higher UTI proportions in selected populations, for example, one HF safety analysis reported UTI in 9.08% of SGLT2 inhibitors users versus 7.10% of placebo users [8]. Reported cases of UTIs range in severity from uncomplicated cystitis to more serious infections such as pyelonephritis and urosepsis [3].

In low- and middle-income countries, including Tanzania, UTI management may be further hindered by limited access to diagnostic microbiology services and widespread empirical antibiotic prescribing, which contributes to the rising prevalence of antimicrobial resistance. However, data describing UTI prevalence and antimicrobial susceptibility specifically among heart failure patients receiving SGLT2 inhibitors in such settings are limited.

Therefore, this study aimed to determine the prevalence of urinary tract infections, associated risk factors, and antimicrobial susceptibility patterns of uropathogens among heart failure patients receiving SGLT2 inhibitors at a tertiary cardiovascular institute in Tanzania.

## Methods

### Study design and setting

This hospital-based cross-sectional study was conducted at Jakaya Kikwete Cardiac Institute (JKCI), Dar es Salaam, Tanzania, from 1st March to 30th June 2024. JKCI is a national referral center mainly offering cardiovascular care in Tanzania and receives a large and diverse population of heart failure patients from across the country and other neighboring countries. This makes it a strategic and representative setting for studying the burden of UTIs in patients receiving SGLT2 inhibitors.

### Study population

The study population comprised heart failure patients aged 18 years and above, who were eligible heart failure patients who received SGLT2 inhibitors (i.e., either Empagliflozin or Dapagliflozin) for at least two weeks and attended JKCI during the study period. Patients who were on antibiotic treatment within the last seven days or those with urethral catheters were excluded from the study.

### Sample size and sampling technique

Based on the available recent literature, the prevalence of UTI among heart failure patients on SLGT2i is approximately 9.08% [4]. The sample size formula for estimating a single proportion was used [9].

$$n = (Z^2 * p * (1 - p))/ (E^2)$$

Where:

- n is the required sample size

- Z is the Z-score corresponding to the desired confidence level (e.g., Z = 1.96 for a 95% confidence level)

- p is the expected prevalence of urinary tract infections

- E is the desired level of precision (margin of error)

This resulted in a minimum sample size of 128 participants; after adjusting for a 10% non-response rate, 141 participants were enrolled consecutively.

### Ethical approval and consent to participate

The MUHAS Institutional Review Board granted ethical approval to conduct this study. Permission to perform this study was also obtained from JKCI and Muhimbili National Hospital for laboratory investigations at the Central Pathology

 

Laboratory (CPL). Consent to participate in this study was obtained from the patient by signing a written informed consent form. This study caused no harm to the study participants. For the participants who had positive UTI tests, results were communicated to the primary care physician for management, and all other participants' information was confidential.

## Data and specimen collection

Using a structured questionnaire, data on socio-demographic characteristics, medical history, and medication details were recorded and verified through review of patients' electronic medical records and pharmacy dispensing logs to confirm accuracy. Participants were then instructed on how to provide a clean-catch midstream urine sample. A clean voided 10–20 milliliters of midstream urine specimen were collected in a sterile wide screw-capped container and transported within 10 minutes in a cool box to the Central Pathology Laboratory (CPL) at the microbiology department for culture and sensitivity testing.

## Laboratory analysis

**Urine culture and identification of uropathogens.** Urinalysis procedures were performed before culture, both the dipstick (assessing leukocytes, nitrites, protein, and glucose) and urine microscopy (evaluating pyuria, bacteriuria, and presence of yeast cells or epithelial cells). A positive urinalysis was defined as presence of leukocyte esterase and/or nitrites and ≥10 WBC/HPF on microscopy. Although urinalysis findings were used as a screening tool, urine culture remained the reference standard for UTI diagnosis, and only culture confirmed cases were included in the final analysis. The urine sample was inoculated using a calibrated 1 microliter disposable loop onto a Cysteine, lactose electrolyte-deficient (CLED) agar used for colony enumeration and blood agar media used for morphological identification and contamination criteria. After incubation at 35–37 °C for 24 hours, significant growth is defined as ≥$10^5$ CFU/ml (or ≥$10^4$ CFU/ml in symptomatic patients that also proceeded to bacterial identification and susceptibility testing). Mixed growth of more than two species was considered contamination and excluded from analysis.

Following incubation, bacterial growth was examined, and uropathogens were identified based on colony morphology, Gram staining, and standard biochemical tests. Gram-negative bacteria were identified using tests such as triple sugar iron (TSI), indole, citrate, and urease, while gram-positive bacteria were identified using catalase and coagulase tests [10], while the *Candida* spp. were identified by colony appearance on germ tube testing.

**Antimicrobial susceptibility testing.** Antimicrobial susceptibility testing (AST) was conducted using the Kirby-Bauer disk diffusion method following CLSI M100 (2024) performance standards and guidelines [11].

Bacterial suspensions were prepared by emulsifying 3–5 well isolated colonies into a tube containing 2–5 ml sterile normal saline (0.9% NaCl) and mixing gently until a homogeneous suspension formed. The suspension's turbidity was then adjusted to 0.5 McFarland standards to standardize the inoculum size. A sterile cotton swab was then dipped into the suspension and swabbed on the surface of the Mueller-Hinton agar plate. Antimicrobial panels were selected according to the gram reaction of the isolated organism and CLSI M100 (2024) recommendations, therefore gram-negative and gram- positive bacteria were tested using antibiotic panels appropriate for each bacterial group. Standard antimicrobial-impregnated discs based on the uropathogen isolated were then aseptically placed on the culture.

The inoculated agar plates were incubated at 37 °C for 18–24 hours. The diameter of the zones of inhibition was measured in millimeters. The interpretation breakpoints were based on whether the bacteria were Susceptible (S), Intermediate (I), or resistant (R) to the tested antibiotic according to CLSI M100 (2024) recommendations [11]. Based on the type of isolated microorganism and CLSI recommendations, commercially prepared antibiotic disks (Oxoid™, Thermo Fisher Scientific, Basingstoke, UK), the antibiotics discs that were used with their respective concentrations include Amikacin 30 µg (AK), Ampicillin 10 µg (AMP), Amoxy-clavulanate 20/10 µg (AMC), Cefepime 30 µg (FEP), Ceftazidime 30 µg (CAZ), Ceftriaxone 30 µg (CRO), Cefoxitin 30 µg (FOX), Ciprofloxacin 5 µg (CIP), Gentamicin 10 µg (GEN), Meropenem 10 µg (MEM), Nalidixic acid 30 µg (NA), Nitrofurantoin 300 µg (F), Penicillin G 10 units (PEN G), Piperacillin-tazobactam 100/10 µg (TZP), Trimethoprim-sulfamethoxazole 1.25/23.75 µg (SXT), Tetracycline 30 µg (TE), Tobramycin 10 µg (TOB).

### Data analysis

Data was analyzed using SPSS v26. Categorical variables were summarized as frequencies (%) and continuous variables as medians (IQR). Associations with UTI were evaluated using modified Poisson regression with robust variance, reporting crude (cPR) and adjusted prevalence ratios (aPR) with 95% CIs. Covariates were selected a priori and by bivariable screening (p ≤ 0.20) with attention to model parsimony; a two-sided p < 0.05 denoted statistical significance.

## Results

### Socio-demographic and clinical characteristics

A total of 141 participants were enrolled in the study, of whom 138 provided viable urine samples for analysis. Most participants were males (55.3%) and over half (53.9%) were aged ≤60 years. Most of the study participants (56%) were outpatients. The use of SGLT2 inhibitors was even between Empagliflozin 10 mg (53.2%) and Dapagliflozin 10 mg (46.8%). SGLT2 inhibitor use was predominantly less than 4 months (87%), **Table 1** and **Table 2**.

### Prevalence of UTIs

A total of 141 HF patients on SGLT2 inhibitors were enrolled in this study. Of these,138 patients provided viable urine samples for culture analysis. Three samples were excluded due to contamination and therefore omitted from further microbiological evaluation.

Among the 138 urine cultures processed,22 demonstrated significant growth, resulting in a UTI prevalence of 15.9% in this population.

### Risk factors

Advanced age (>60 years) (adjusted prevalence ratio [aPR] 3.77, 95% CI: 1.42–10.01), female sex (aPR 2.92, 95% CI: 1.19–7.15), and longer duration of SGLT2 inhibitor use (≥4 months) (aPR 3.19, 95% CI: 1.70–5.96) were significantly associated with increased UTI risk (Table 3).

### Uropathogens identified

A total of 22 uropathogens were isolated from the 138 urine cultures processed. No polymicrobial growth was observed, as each culture-positive sample yielded a single significant organism (≥ $10^5$ CFU/ml). The predominant uropathogen was *Escherichia coli* (50%, n = 11), followed by *Klebsiella pneumoniae* (13.6%, n = 3), *Pseudomonas aeruginosa* (9.1%, n = 2), *Candida* spp. (9.1%, n = 2), *Staphylococcus aureus* (4.5%, n = 1), and *Staphylococcus saprophyticus* (4.5%, n = 1) Fig 1. No clear trend in uropathogen distribution was observed between empagliflozin and dapagliflozin users. *Escherichia coli* remained the predominant isolate across both SGLT2 inhibitor groups, with no apparent clustering of specific pathogens by drug type. Similarly, uropathogen distribution did not differ between inpatients and outpatients.

### Antimicrobial susceptibility pattern

High resistance was observed against Ampicillin (90.9%), Tetracycline (69.2%), and Ceftazidime (53.8%). In contrast, uropathogens were highly susceptible to Nitrofurantoin (84.6%) and Meropenem (100%). These patterns are summarized in Tables 4 and 5.

## Discussion

This study aimed to assess the prevalence of UTIs associated risk factors and antimicrobial susceptibility patterns of uropathogens isolated from heart failure patients on SGLT2 inhibitors. The study findings indicate a UTI prevalence of 15.9% in this patient population.

**Table 1. Socio-demographic characteristics of the study participants.**

| Variables | | Frequency (n) | Percent (%) |
|---|---|---|---|
| **Age** (Years) | | | |
| | ≤60 | 76 | 53.9 |
| | >60 | 65 | 46.1 |
| **Sex** | | | |
| | Male | 78 | 55.3 |
| | Female | 63 | 44.7 |
| **Menopausal status** | | | |
| | Pre-menopausal | 35 | 55.6 |
| | Post-menopausal | 28 | 44.4 |
| **Marital status** | | | |
| | Married | 99 | 70.2 |
| | Widow/Widower | 26 | 18.4 |
| | Cohabiting | 6 | 4.3 |
| | Single | 5 | 3.5 |
| | Divorced | 5 | 3.5 |
| **Occupational status** | | | |
| | Self-employed | 79 | 56.0 |
| | Retired | 37 | 26.2 |
| | Government/Private employed | 22 | 15.6 |
| | Unemployed | 2 | 1.4 |
| | Student | 1 | 0.7 |
| **Bills coverage** | | | |
| | Health insured | 86 | 61.0 |
| | Cash | 55 | 39.0 |
| **Permanent Residence** | | | |
| | Within Dar es Salaam | 75 | 53.2 |
| | From other regions | 61 | 43.3 |
| | From other countries | 5 | 3.5 |
| **Alcohol drinking habits** | | | |
| | Yes | 13 | 9.2 |
| | No | 128 | 90.8 |

This prevalence is higher compared to the meta-analysis of ten randomized controlled trials involving the high-risk cardiac patients that reported an increased trend of UTI among SGLT2 inhibitors compared to the placebo (9.08% vs 7.10%) [4,8]. The difference may reflect real-world clinical conditions in low-resource settings, where environmental and hygiene related factors may influence infection risk.. Furthermore, Tanzania's tropical climate may predispose patients to dehydration and reduced urine output,potentially increasing urine stasis and risk of bacterial proliferation [12,13]. The increased UTI risk observed in both clinical trials and observational settings is biologically plausible and attributed to the SGLT2 inhibitor- induced glucosuria, which creates an environment conducive to bacterial growth [14].

Age older than 60 years was significantly associated with UTI. The most likely contributor to this risk is age-related changes in immune function and urinary tract anatomy and comorbidities [15]. This is similarly reported in previous studies that investigated both diabetic and non-diabetic patients using SGLT2 inhibitors [16,17]. Moreover, sex was also significantly associated with UTIs, female sex being at higher risk; this finding aligns with known anatomical factors, such as the

**Table 2. Clinical characteristics and medication history of the study participants.**

| Variables | | Frequency (n) | Percent (%) |
|---|---|---|---|
| **Patient admitted as** | | | |
| | Outpatient | 79 | 56.0 |
| | Inpatient | 62 | 44.0 |
| **Type of SGLT2 inhibitor used** | | | |
| | Empagliflozin 10 mg | 75 | 53.2 |
| | Dapagliflozin 10 mg | 66 | 46.8 |
| **Duration of SGLT2 inhibitor use** | | | |
| | < 4 months | 123 | 87 |
| | ≥ 4 months | 18 | 12.8 |
| **Patient vitals** | | | |
| | Median SBP (IQR) (mmHg) | 112 (112, 134) | |
| | Median DBP (IQR) (mmHg) | 83 (78, 88) | |
| | Median Pulse rate (IQR) (bpm) | 88 (83, 92) | |
| **Left ventricular ejection fraction (LVEF) %** | | | |
| | ≤ 40 (HFrEF) | 46 | 32.6 |
| | 41-49 (HFmrEF) | 58 | 41.1 |
| | ≥ 50 (HFpEF) | 37 | 26.2 |
| **BMI status Kg/m$^2$** | | | |
| | Underweight (≤18.5) | 8 | 5.7 |
| | Normal weight (18.5–24.9) | 57 | 40.4 |
| | Overweight (25.0–29.9) | 44 | 31.2 |
| | Obese ≥30 | 32 | 22.7 |
| **New York Heart Association Class (NYHA)** | | | |
| | I&II | 100 | 70.9 |
| | III | 41 | 29.1 |
| **Other Chronic Diseases** | | | |
| | Yes | 83 | 58.9 |
| | No | 58 | 41.1 |
| **Other Chronic Diseases** | | | |
| | Other Cardiovascular diseases | 53 | 63.9 |
| | Diabetes Mellitus | 16 | 19.3 |
| | HIV/AIDS | 3 | 3.6 |
| | Diabetes and other cardiovascular diseases | 8 | 9.6 |
| | HIV/AIDS and other cardiovascular diseases | 3 | 3.6 |
| **Previous history of UTI** | | | |
| | Yes | 18 | 12.8 |
| | No | 86 | 61.0 |
| | Unknown | 36 | 26.2 |
| **UTI symptoms** | | | |
| | Asymptomatic | 116 | 82.3 |
| | Symptomatic | 25 | 17.7 |

**Table 3. Univariable and multivariable analysis of the factors associated with urinary tract infection.**

| Variable | Category | Univariable analysis | | | Multivariable analysis | | |
|---|---|---|---|---|---|---|---|
| | | cPR | 95% CI | p-value | aPR | 95% CI | p-value |
| **Age group** | ≤ 60 | Ref | | | | | |
| | >60 | 5.36 | 1.91–15.01 | 0.001 | 3.77 | 1.42–10.01 | **0.008** |
| **Sex** | Male | Ref | | | | | |
| | Female | 4.05 | 1.58–10.35 | 0.004 | 2.92 | 1.19–7.15 | **0.019** |
| **Admitted as** | Outpatient | 1.11 | 0.51–2.43 | 0.791 | | | |
| | Inpatient | Ref | | | | | |
| **SGLT2 inhibitor** | Empagliflozin 10 mg | Ref | | | | | |
| | Dapagliflozin 10 mg | 0.66 | 0.30–1.47 | 0.311 | | | |
| **Duration of SGLT2i use** | < 4 months | Ref | | | | | |
| | ≥ 4 Months | 6.67 | 3.40–13.07 | **< 0.001** | 3.19 | 1.70–5.96 | **< 0.001** |
| **Chronic diseases** | Yes | 1.55 | 0.68–3.57 | 0.299 | | | |
| | No | Ref | | | | | |
| **DM** | Yes | 1.47 | 0.60–3.59 | 0.396 | | | |
| | No | Ref | | | | | |
| **History of UTI** | Yes | 3.78 | 1.74–8.23 | **0.001** | 1.58 | 0.90–2.79 | 0.113 |
| | Unknown | 0.97 | 0.33–2.89 | 0.958 | 0.85 | 0.37–1.99 | 0.713 |
| | No | Ref | | | | | |

Key: cPR: crude Prevalence Ratio, aPR: adjusted Prevalence Ratio, Ref: Reference, DM: Diabetes mellitus, NYHA: New York Heart Association, BMI: Body Mass Index

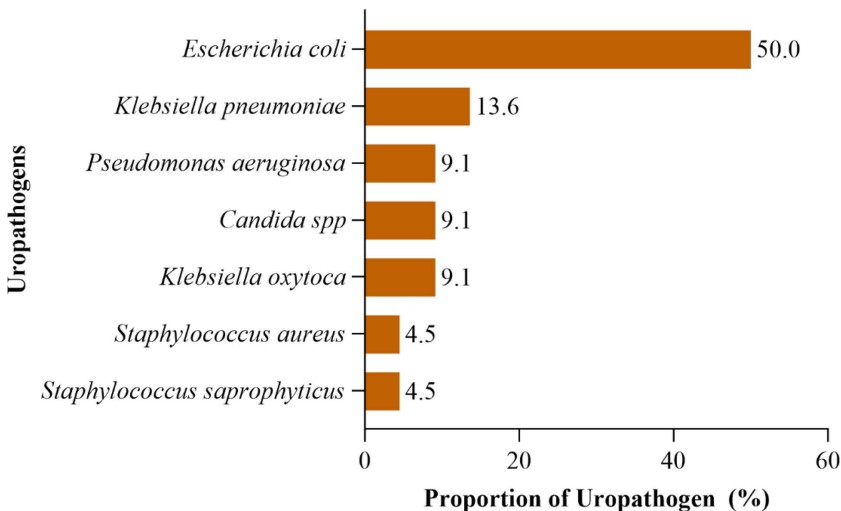

**Fig 1. Proportion of isolated uropathogens among study participants' inhibitors with a positive culture.**

urethra being closer to the anus and the urethral opening being near the bladder, together with the hormonal factors that predispose women to UTI more than men [18].

Furthermore, the duration of SGLT2 inhibitor use was also a significant factor for UTIs among patients; those who had been on the medication for four months had a markedly higher risk of UTIs. This is due to persistent glucosuria, which

**Table 4. Antimicrobial resistance pattern of gram-negative bacteria isolated from urine culture.**

| Bacterial Isolates | n | Antimicrobial agents resisted (%) | | | | | | | | | | | | | | |
|---|---|---|---|---|---|---|---|---|---|---|---|---|---|---|---|---|
| | | AK | AMP | AMC | FEP | CAZ | CRO | CIP | GEN | MEM | NA | F | TZP | SXT | TE | TOB |
| *Escherichia Coli* | 11 | 27.3 | 90.9 | 45.5 | 54.5 | 70.0 | 54.5 | 36.4 | 36.4 | 0 | 54.5 | 0 | 30.0 | 54.5 | 40.0 | – |
| *Klebsiella pneumoniae* | 3 | 33.3 | 100 | 33.3 | 33.3 | 33.3 | 33.3 | 33.3 | 33.3 | 0 | 33.3 | 33.3 | 0 | 66.7 | 66.7 | – |
| *Pseudomonas aeruginosa* | 2 | 100 | – | – | 0 | 100 | – | 0 | – | 50.0 | – | – | 100 | – | – | 0 |
| *Klebsiella oxytoca* | 2 | 0 | 100 | 50.0 | 0 | 0 | 0 | 0 | 50.0 | 0 | 0 | 50.0 | 0 | 50.0 | 0 | – |

Key: (-) Not applicable, AK: Amikacin, AMP: Ampicillin, AMC: Amoxicillin-Clavulanic acid, FEP: Cefepime, CAZ: Ceftazidime, CRO: Ceftriaxone, CIP: Ciprofloxacin, GEN: Gentamycin, MEM: Meropenem, NA: Nalidixic acid, F: Nitrofurantoin, TZP: Piperacillin/tazobactam
SXT: Trimethoprim/Sulfamethoxazole, TE: Tetracycline, TOB: Tobramycin.

**Table 5. Antimicrobial resistance pattern of gram-positive bacteria isolated from urine culture.**

| Bacterial Isolates | n | Antimicrobial agents resisted (%) | | | | | | |
|---|---|---|---|---|---|---|---|---|
| | | FOX | CIP | GE | F | Pen G | SXT | TE |
| *Staphylococcus aureus* | 1 | 0 | 100 | 0 | 0 | 0 | 0 | 100 |
| *Staphylococcus saprophyticus* | 1 | – | 0 | 100 | 0 | 100 | 100 | 100 |

Key:(-) Not applicable, FOX: Cefoxitin, CIP: Ciprofloxacin, GEN: Gentamycin, F: Nitrofurantoin, Pen G: Penicillin G, SXT: Sulfamethoxazole-trimethoprim, TE: Tetracycline.

increases the risk of bacterial proliferation and subsequent infection [19]. This finding is consistent with a study that examined the incidence of UTIs among patients on SGLT2 inhibitors, which reported that most UTI episodes occurred between three and six months of treatment [20]. While SGLT2 inhibitors initially causes osmotic diuresis and polyuria, this effect generally stabilizes after the early treatment phase. Glucosuria, however, persists throughout the therapy. Therefore, the increased risk observed after prolong ed use may reflect the cumulative exposure to glucose rich urine rather than transient early polyuria [21]. In this study, there was no statistical difference in the risk of UTIs that was observed in patients taking either Dapagliflozin 10 mg or Empagliflozin. This finding suggests a possible class effect rather than a drug specific risk; however, some studies have reported varying risk profiles with an increased risk of UTI with Dapagliflozin compared to Empagliflozin [22].

The study identified several uropathogens responsible for UTIs in this population, with *Escherichia coli* being the most frequently isolated pathogen (50%). This finding is consistent with global trends, where *E. coli* is the predominant uropathogen in both community-acquired and healthcare-associated UTIs [23]. *Klebsiella pneumoniae* and *Pseudomonas aeruginosa* were also commonly isolated, consistent with findings from studies conducted among patients receiving SGLT2 inhibitors in both low- and middle-income settings [24]. *Staphylococcus aureus* and *Staphylococcus saprophyticus* were responsible for the infections of Gram-positive organisms. The isolation of Candida spp. may be associated with glycosuria-induced fungal overgrowth, which has been previously linked to SGLT2 inhibitor use [20,24]. The diversity of pathogens highlights the need for comprehensive diagnostic approaches to identify the causative organisms accurately and guide effective treatment.

Based on the isolated uropathogens, high resistance was noted with 100% and 90.9% of Klebsiella spp and *Escherichia coli* being resistant to Ampicillin like other studies in Tanzania [25,26]. This persistent absolute resistance of Ampicillin even to other East African countries has led to the revocation of this product as a monotherapy in Tanzania by TMDA [27]. The least resistance to no resistance was observed in Meropenem and Ciprofloxacin. *P. aeruginosa*, an alert micro-organism, has shown resistance to Amikacin, Piperacillin/tazobactam and is susceptible to Ciprofloxacin. The

gram-positive bacteria are more resistant to tetracycline. *Staphylococcus saprophyticus* is resistant to gentamycin, penicillin G, and Sulfamethoxazole/trimethoprim, that is susceptible to *Staphylococcus aureus*, these gram-positive bacteria have shown a high susceptibility to nitrofurantoin which is also among the first-line treatment of UTIs. This resistance pattern is consistent with other studies [28–30].

The observed resistance patterns underscore the importance of routine urine culture and antimicrobial susceptibility testing, particularly among HF patients receiving SGLT2 inhibitors, to guide targeted therapy and support antimicrobial stewardship efforts.

### Study limitations

This cross-sectional design limits causality or the temporal relationship between SGLT2 inhibitor initiation and UTI development. In addition, the study was conducted at a single specialized cardiac centre, which may limit generalizability. We were unable to assess the baseline UTI status before initiation of SGLT2 inhibitor therapy, which restricts the ability to fully attribute UTI occurrence to medication exposure alone. Moreover, multidrug resistance (MDR) criteria was not predefined as analytical outcome, and a formal MDR categorization using internationally standardized criteria was therefore not performed, this limits the ability to comprehensively quantify the burden of MDR uropathogens in this population.. The relatively small number of positive isolates (n = 22) also limited the power to conduct more detailed subgroup analyses. Despite these limitations, the use of standardized semi-quantitative urine culture methods and CLSI-based antimicrobial susceptibility testing strengthens the validity of our findings.

### Conclusion

This study demonstrated that UTIs are prevalent among heart failure patients on SGLT2 inhibitors, with *Escherichia coli* being the most isolated pathogen. A notable resistance to commonly prescribed antibiotics indicates the need for culture-based diagnosis and sensitivity testing before initiating therapy. Furthermore, these findings suggest the need for an evidence-based decision in selecting empirical treatment for UTIs and thereby promoting the rational use of antibiotics.

This study further underscores the importance of UTI surveillance in HF patients on SGLT2 inhibitors, particularly in high-risk groups such as older adults and women, especially in resource-limited settings. Targeted antimicrobial therapy and preventive measures are essential to improving patient outcomes and addressing AMR challenges in Tanzania.

### Supporting information

**S1 File. Julieth Working file rev (1).**
(SAV)

### Acknowledgments

We would like to extend our gratitude to the Jakaya Kikwete Cardiac Institute (JKCI) for granting permission and providing support to conduct this study. Special thanks go to the microbiology staff members at CPL and other departments for their assistance in sample collection, culture, and testing procedures. We also appreciate the contributions of the Muhimbili University of Health and Allied Sciences (MUHAS) for their guidance throughout this research. Finally, we acknowledge the patients who participated in this study, without whom this research would not have been possible.

### Author contributions

**Conceptualization:** Julieth Kaaya, Deus Buma, Hamu Mlyuka, Samwel Rweyemamu, Raphael Sangeda, Ritah F Mutagonda.

**Data curation:** Julieth Kaaya, Deus Buma, Hamu Mlyuka, Ritah F Mutagonda.

**Formal analysis:** Julieth Kaaya, Albert Ntukula, Raphael Sangeda, Ritah F Mutagonda.

**Funding acquisition:** Julieth Kaaya.

**Investigation:** Julieth Kaaya, Deus Buma, Albert Ntukula, Helfrid B. Ilomo.

**Methodology:** Julieth Kaaya, Deus Buma, Samwel Rweyemamu, Elias Bukundi, Raphael Sangeda, Ritah F Mutagonda.

**Project administration:** Julieth Kaaya.

**Resources:** Deus Buma, Albert Ntukula.

**Software:** Helfrid B. Ilomo.

**Supervision:** Deus Buma, Hamu Mlyuka, Elias Bukundi, Raphael Sangeda, Ritah F Mutagonda.

**Validation:** Martine A. Manguzu, Hamu Mlyuka, Raphael Sangeda, Ritah F Mutagonda.

**Visualization:** Martine A. Manguzu, Deus Buma, Hamu Mlyuka, Ritah F Mutagonda.

**Writing – original draft:** Julieth Kaaya, Martine A. Manguzu.

**Writing – review & editing:** Deus Buma, Hamu Mlyuka, Samwel Rweyemamu, Albert Ntukula, Helfrid B. Ilomo, Elias Bukundi, Raphael Sangeda, Ritah F Mutagonda.

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
