## [Decision Letter · Decision Letter 0]

3 Oct 2025

Urinary Tract Infections and Drug Resistance Patterns in Heart Failure Patients on Sodium-Glucose Transporter 2 Inhibitors: Evidence from Jakaya Kikwete Cardiac Institute in Tanzania

PLOS ONE

Dear Dr. Mutagonda,

Thank you for submitting your manuscript to PLOS ONE. After careful consideration, we feel that it has merit but does not fully meet PLOS ONE’s publication criteria as it currently stands. Therefore, we invite you to submit a revised version of the manuscript that addresses the points raised during the review process.

We look forward to receiving your revised manuscript.

Kind regards,

Tombari Pius Monsi, Ph.D

Academic Editor

PLOS ONE

Journal Requirements:

2. Please update your submission to use the PLOS LaTeX template. The template and more information on our requirements for LaTeX submissions can be found at http://journals.plos.org/plosone/s/latex ..

5. We note you have included a table to which you do not refer in the text of your manuscript. Please ensure that you refer to Table 3 in your text; if accepted, production will need this reference to link the reader to the Table.

Reviewers' comments:

Reviewer's Responses to Questions

**Comments to the Author**

1. Is the manuscript technically sound, and do the data support the conclusions?

Reviewer #1: No

Reviewer #2: Partly

2. Has the statistical analysis been performed appropriately and rigorously?

Reviewer #1: No

Reviewer #2: Yes

3. Have the authors made all data underlying the findings in their manuscript fully available?

Reviewer #1: No

Reviewer #2: Yes

4. Is the manuscript presented in an intelligible fashion and written in standard English?

Reviewer #1: Yes

Reviewer #2: No

Reviewer #1: The title needs to be changed to capture the risk factors that have extensively been discussed in the study. The author also needs to go through the article attached as it has additional comments on how to make the manuscript better.

Reviewer #2: Title

The title is clear, well structured, and easy to understand.

Abstract

• Replace “midstream urine samples” with midstream clean catch urine samples throughout the text whenever applicable.

• Use the correct term for urine culture semi-quantitative urine culture, consistently.

• Revise the sentence: “Out of 138 participants among those cultured”—this is misleading, as patients themselves are not cultured; rather, their biological samples (e.g., urine) are cultured.

• Ensure all percentages are reported with one decimal place (e.g., 50.0%).

• In the conclusion, specify the study population to emphasize clinical application, e.g., “… routine urine culture and sensitivity testing among (HF patients on SGLT2 inhibitors) to guide appropriate therapy …”.

Background

• Support the second paragraph with proportions or prevalence rates of UTI in HF patients on SGLT2 inhibitors compared to those not on SGLT2 inhibitors, including contradictory findings. This will strengthen the justification for the current study.

Methods

• Study population: delete the repeated word “receiving.”

• Sample size estimation: provide a citation for the reference used in sample size calculation.

• Clearly state whether the urine collection containers were sterile.

• Laboratory analysis: revise the subheading, as the words “culture” and “isolation” convey the same concept.

• Delete repeated wording in: “The urine sample was inoculated (at the microbiology laboratory) using a calibrated 1 microliter disposable loop in a cysteine, lactose electrolyte-deficient (CLED) agar, and blood agar media”, since the laboratory has already been mentioned earlier.

• Define “significant growth” in the sentence: “After 24 hours of incubation at 35–37 °C, significant growth (≥105 CFU/ml) was further analyzed.”

• Delete “isolated” from: “… and specific bacteria were (isolated) and identified through colony morphology, gram reaction, and biochemical tests.”

• Report bacterial growth using SI units: >10⁴ CFU/ml.

• In case of pure growth with bacterial quantity of ≥104 CFU/ml, what was the way forward?

• Revise: “Bacterial isolation and identification were conducted using standard culture and biochemical tests” by briefly describing the tests used for Gram-negative and Gram-positive bacteria, and citing an appropriate reference.

• Revise: “Urinalysis preceded urine culture on CLED and blood agar, with uropathogens identified by colony morphology and biochemical tests” to clearly describe the urinalysis procedure/test performed e.g., microscopy or dipstick. Additionally, respective results should be provided in the Results sections and discussed in the Discussion section.

• Cite an appropriate reference for the Kirby Bauer disk diffusion method.

• Antimicrobial susceptibility testing: delete the word “one,” which may be misleading for organisms with small colony size such as Enterococcus spp or Streptococcus spp.

• Include brand, city, and country of manufacturer for all culture media and antimicrobial discs.

• Provide the abbreviation for Penicillin G.

Results

• Begin the section on sociodemographic and clinical characteristics with the total number of participants enrolled.

• Prevalence of UTI and risk factors: do not start sentences with numerals. Clarify why 138 urine samples were processed out of 141 patients. The 3 missing samples, whether not submitted to the lab or rejected by the lab, should be excluded from analysis.

• Uropathogens identified: do not italicize “spp” or “species” (see: Candida spp (9.1%)).

• State the total number of uropathogens identified from the 138 urine cultures. Clarify whether any samples yielded 2 significant organisms (≥10⁵ CFU/ml), resulting in >138 isolates.

• Provide actual counts in the prevalence statement, e.g., Escherichia coli (50.0%; n=69) or (50.0%; 69/138).

• Table 4: define “(TM)” in the footnotes and arrange all abbreviations and their definitions in alphabetical order.

Discussion

• In the first paragraph, include prevalence estimates reported in relevant meta-analyses, with citations.

• Correct all typographical errors, such as missing full stops or spaces between sentences (see first paragraph).

• The following sentence is not clear: “The observed late occurrence of UTI events of up to three months can be explained by initial polyuria caused by the SGLT2 inhibitors …”. Clarify whether polyuria decreases after three months of using SGLT2, and whether glucose levels during initial polyuria (first 3 months following the use of SGLT2) are insufficient for microbial colonization and proliferation.

• Explain the clinical implications of your findings for the management of HF patients on SGLT2.

• Add a Study limitations section following the Discussion and before the Conclusion.

References

• Cite the appropriate CLSI as suggested on their guidelines page no. ii (Suggested citation).

(what does this mean? ). If published, this will include your full peer review and any attached files.). If published, this will include your full peer review and any attached files.

**Do you want your identity to be public for this peer review?**  For information about this choice, including consent withdrawal, please see our  For information about this choice, including consent withdrawal, please see our Privacy Policy .

Reviewer #1: No

Reviewer #2: No

While revising your submission, please upload your figure files to the Preflight Analysis and Conversion Engine (PACE) digital diagnostic tool, https://pacev2.apexcovantage.com/ . PACE helps ensure that figures meet PLOS requirements. To use PACE, you must first register as a user. Registration is free. Then, login and navigate to the UPLOAD tab, where you will find detailed instructions on how to use the tool. If you encounter any issues or have any questions when using PACE, please email PLOS at . PACE helps ensure that figures meet PLOS requirements. To use PACE, you must first register as a user. Registration is free. Then, login and navigate to the UPLOAD tab, where you will find detailed instructions on how to use the tool. If you encounter any issues or have any questions when using PACE, please email PLOS at figures@plos.org . Please note that Supporting Information files do not need this step.. Please note that Supporting Information files do not need this step.

---

## [Author Response · Author response to Decision Letter 1]

25 Nov 2025

The response to reviewers document has been attached

---

## [Editor Report · Decision Letter 1]

28 Dec 2025

Dear Dr. Mutagonda,

**1. Antimicrobial Susceptibility Testing in the Method section. Revisit this section****2. The Result section: You attempted to correct some errors here but not all****3. The Discussion section. You skipped some of the corrections here**

Kind regards,

Tombari Pius Monsi, Ph.D

Academic Editor

PLOS One

---

## [Author Response · Author response to Decision Letter 2]

25 Feb 2026

Apologies for overlooking the comments that were submitted in soft copy. The remaining responses to the reviewers’ comments have now been attached.

---

## [Editor Report · Decision Letter 2]

2 Mar 2026

Urinary tract infections, Risk factors and antimicrobial Resistance Patterns in Heart Failure patients on Sodium-Glucose Transporter 2 Inhibitors: Evidence from Jakaya Kikwete Cardiac Institute in Tanzania

PONE-D-25-48414R2

Dear Dr. Mutagonda,

We’re pleased to inform you that your manuscript has been judged scientifically suitable for publication and will be formally accepted for publication once it meets all outstanding technical requirements.

Kind regards,

Tombari Pius Monsi, Ph.D

Academic Editor

PLOS One

---

## [Editor Report · Acceptance letter]

PONE-D-25-48414R2

PLOS One

Dear Dr. Mutagonda,

I'm pleased to inform you that your manuscript has been deemed suitable for publication in PLOS One. Congratulations! Your manuscript is now being handed over to our production team.

Kind regards,

on behalf of

Dr. Tombari Pius Monsi

Academic Editor

PLOS One